# Quantification of glucose-6-phosphate dehydrogenase activity by spectrophotometry: A systematic review and meta-analysis

Daniel A. Pfeffer[1]*, Benedikt Ley[1], Rosalind E. Howes[2,3], Patrick Adu[4], Mohammad Shafiul Alam[5], Pooja Bansil[6], Yap Boum, II[7,8], Marcelo Brito[9], Pimlak Charoenkwan[10], Archie Clements[11,12], Liwang Cui[13], Zeshuai Deng[14], Ochaka Julie Egesie[15], Fe Esperanza Espino[16], Michael E. von Fricken[17], Muzamil Mahdi Abdel Hamid[18], Yongshu He[14], Gisela Henriques[19], Wasif Ali Khan[5], Nimol Khim[20], Saorin Kim[20], Marcus Lacerda[9], Chanthap Lon[21], Asrat Hailu Mekuria[22], Didier Menard[23], Wuelton Monteiro[9], François Nosten[24,25], Nwe Nwe Oo[26], Sampa Pal[6], Duangdao Palasuwan[27], Sunil Parikh[28], Ayodhia Pitaloka Pasaribu[29], Jeanne Rini Poespoprodjo[30], David J. Price[31,32], Arantxa Roca-Feltrer[33], Michelle E. Roh[34], David L. Saunders[21,35,36], Michele D. Spring[21], Inge Sutanto[37], Kamala Ley-Thriemer[1], Thomas A. Weppelmann[38], Lorenz von Seidlein[24,39], Ari Winasti Satyagraha[40], Germana Bancone[24,25], Gonzalo J. Domingo[6], Ric N. Price[1,25,39]

**Data Availability Statement:** All relevant aggregate-level data are included in the manuscript and Supporting information. Interested individuals are encouraged to contact authors of included

1 Global and Tropical Health Division, Menzies School of Health Research and Charles Darwin University, Darwin, Australia, 2 Malaria Atlas Project, Big Data Institute, Nuffield Department of Medicine, University of Oxford, Oxford, United Kingdom, 3 Foundation for Innovative New Diagnostics, Geneva, Switzerland, 4 Department of Medical Laboratory Sciences, School of Allied Health Sciences, University of Cape Coast, Cape Coast, Ghana, 5 Infectious Diseases Division, International Centre for Diarrheal Diseases Research, Bangladesh, Mohakhali, Dhaka, Bangladesh, 6 Diagnostics Program, PATH, Seattle, Washington, United States of America, 7 Médecins sans Frontières Epicentre, Mbarara Research Centre, Mbarara, Uganda, 8 Mbarara University of Science and Technology, Mbarara, Uganda, 9 Fundação de Medicina Tropical Dr. Heitor Vieira Dourado, Manaus, Amazonas, Brasil, 10 Division of Hematology and Oncology, Department of Pediatrics, Faculty of Medicine, Chiang Mai University, Chiang Mai, Thailand, 11 Faculty of Health Sciences, Curtin University, Bentley, Australia, 12 Telethon Kids Institute, Nedlands, Australia, 13 Department of Entomology, Pennsylvania State University, University Park, Pennsylvania, United States of America, 14 Department of Cell Biology and Medical Genetics, Kunming Medical University, Kunming, Yunnan Province, China, 15 Department of Hematology and Blood Transfusion, Faculty of Medical Sciences, University of Jos and Jos University Teaching Hospital, Jos, Plateau State, Nigeria, 16 Department of Parasitology, Research Institute for Tropical Medicine, Department of Health, Alabang, Muntinlupa City, Philippines, 17 Department of Global and Community Health, George Mason University, Fairfax, Virginia, United States of America, 18 Department of Parasitology and Medical Entomology, Institute of Endemic Diseases, University of Khartoum, Khartoum, Republic of the Sudan, 19 Faculty of Infectious and Tropical Diseases, London School of Hygiene & Tropical Medicine, London, United Kingdom, 20 Malaria Molecular Epidemiology Unit, Institut Pasteur du Cambodge, Phnom Penh, Cambodia, 21 Armed Forces Research Institute of Medical Sciences, Bangkok, Thailand, 22 School of Medicine, Addis Ababa University, Addis Ababa, Ethiopia, 23 Malaria Genetics and Resistance Group, Institut Pasteur, Paris, France, 24 Shoklo Malaria Research Unit, Mahidol–Oxford Tropical Medicine Research Unit, Faculty of Tropical Medicine, Mahidol University, Mae Sot, Thailand, 25 Centre for Tropical Medicine & Global Health, Nuffield Department of Medicine, University of Oxford, Oxford, United Kingdom, 26 Department of Medical Research, Lower Myanmar, Yangon, Myanmar, 27 Oxidation in Red Cell Disorders and Health Research Unit, Department of Clinical Microscopy, Faculty of Allied Health Sciences, Chulalongkorn University, Bangkok, Thailand, 28 Yale School of Public Health, New Haven, Connecticut, United States of America, 29 Faculty of Medicine, Universitas Sumatera Utara, Medan, Indonesia, 30 Yayasan Pengembangan Kesehatan dan Masyarakat Papua (YPKMP), Papua, Indonesia, 31 Centre for Epidemiology and Biostatistics, Melbourne School of Population and Global Health, University of Melbourne, Melbourne, Australia, 32 The Peter Doherty Institute for Infection and Immunity, The University of Melbourne and Royal Melbourne Hospital, Melbourne, Australia, 33 Malaria Consortium, Phnom Penh, Cambodia, 34 Global Health Group, Malaria Elimination Initiative, University of California, San Francisco, San Francisco, United States of America, 35 F. Edward Hebert

primary studies to obtain individual-level data (contact details are in S4 File).

**Funding:** This work was funded by the Wellcome Trust (200909 awarded to RNP) and the Bill & Melinda Gates Foundation (OPP1164105). GB and FN are part of the Wellcome Trust Mahidol University Oxford Tropical Medicine Research Programme funded by the Wellcome Trust. This work was supported by the Australian Centre for Research Excellence on Malaria Elimination (ACREME), funded by the National Health and Medical Research Council of Australia (APP 1134989).

**Abbreviations:** AMM, adjusted male median; CV, coefficient of variation; G6PD, glucose-6-phosphate dehydrogenase; G6PDd, G6PD deficient; G6PDn, G6PD normal; NPV, negative predictive value; QUADAS-2, Quality Assessment of Diagnostic Accuracy Studies-2; RBC, red blood cell; UV, ultraviolet; WHO, World Health Organization.

School of Medicine, Uniformed Services University of the Health Sciences, Bethesda, Maryland, United States of America, **36** US Army Medical Materiel Development Activity, Fort Detrick, Maryland, United States of America, **37** University of Indonesia, Jakarta, Indonesia, **38** Herbert Wertheim College of Medicine, Florida International University, Miami, Florida, United States of America, **39** Mahidol Oxford Tropical Medicine Research Unit, Faculty of Tropical Medicine, Mahidol University, Bangkok, Thailand, **40** Eijkman Institute for Molecular Biology, Jakarta, Indonesia

* daniel.pfeffer@menzies.edu.au

# Abstract

## Background

The radical cure of *Plasmodium vivax* and *P. ovale* requires treatment with primaquine or tafenoquine to clear dormant liver stages. Either drug can induce haemolysis in individuals with glucose-6-phosphate dehydrogenase (G6PD) deficiency, necessitating screening. The reference diagnostic method for G6PD activity is ultraviolet (UV) spectrophotometry; however, a universal G6PD activity threshold above which these drugs can be safely administered is not yet defined. Our study aimed to quantify assay-based variation in G6PD spectrophotometry and to explore the diagnostic implications of applying a universal threshold.

## Methods and findings

Individual-level data were pooled from studies that used G6PD spectrophotometry. Studies were identified via PubMed search (25 April 2018) and unpublished contributions from contacted authors (PROSPERO: CRD42019121414). Studies were excluded if they assessed only individuals with known haematological conditions, were family studies, or had insufficient details. Studies of malaria patients were included but analysed separately. Included studies were assessed for risk of bias using an adapted form of the Quality Assessment of Diagnostic Accuracy Studies-2 (QUADAS-2) tool. Repeatability and intra- and interlaboratory variability in G6PD activity measurements were compared between studies and pooled across the dataset. A universal threshold for G6PD deficiency was derived, and its diagnostic performance was compared to site-specific thresholds. Study participants ($n = 15,811$) were aged between 0 and 86 years, and 44.4% (7,083) were women. Median (range) activity of G6PD normal (G6PDn) control samples was 10.0 U/g Hb (6.3–14.0) for the Trinity assay and 8.3 U/g Hb (6.8–15.6) for the Randox assay. G6PD activity distributions varied significantly between studies. For the 13 studies that used the Trinity assay, the adjusted male median (AMM; a standardised metric of 100% G6PD activity) varied from 5.7 to 12.6 U/g Hb ($p < 0.001$). Assay precision varied between laboratories, as assessed by variance in control measurements (from 0.1 to 1.5 U/g Hb; $p < 0.001$) and study-wise mean coefficient of variation (CV) of replicate measures (from 1.6% to 14.9%; $p < 0.001$). A universal threshold of 100% G6PD activity was defined as 9.4 U/g Hb, yielding diagnostic thresholds of 6.6 U/g Hb (70% activity) and 2.8 U/g Hb (30% activity). These thresholds diagnosed individuals with less than 30% G6PD activity with study-wise sensitivity from 89% (95% CI: 81%–94%) to 100% (95% CI: 96%–100%) and specificity from 96% (95% CI: 89%–99%) to 100% (100%–100%). However, when considering intermediate deficiency (<70% G6PD activity), sensitivity fell to a minimum of 64% (95% CI: 52%–75%) and specificity to 35%

(95% CI: 24%–46%). Our ability to identify underlying factors associated with study-level heterogeneity was limited by the lack of availability of covariate data and diverse study contexts and methodologies.

## Conclusions

Our findings indicate that there is substantial variation in G6PD measurements by spectrophotometry between sites. This is likely due to variability in laboratory methods, with possible contribution of unmeasured population factors. While an assay-specific, universal quantitative threshold offers robust diagnosis at the 30% level, inter-study variability impedes performance of universal thresholds at the 70% level. Caution is advised in comparing findings based on absolute G6PD activity measurements across studies. Novel handheld quantitative G6PD diagnostics may allow greater standardisation in the future.

## Author summary

### Why was this study done?

- Complete cure of vivax malaria, the most geographically widespread malaria species, requires the use of 8-aminoquinoline drugs to clear dormant liver stages of the parasite ('radical cure'); however, these drugs can cause severe haemolysis in individuals with glucose-6-phosphate dehydrogenase (G6PD) deficiency.

- Ultraviolet (UV) spectrophotometry is used as the reference test to measure G6PD activity, for validating new point-of-care diagnostics, and to determine population-specific definitions of G6PD deficiency.

- Currently, there is no universal threshold to define G6PD deficiency, and each laboratory must invest time and resources to derive site- and laboratory-specific definitions of G6PD deficiency.

### What did the researchers do and find?

- We pooled measurements of G6PD activity from studies conducted across different countries and laboratories worldwide.

- We assessed the comparability of spectrophotometry results between these laboratories to see whether a universal definition and diagnostic cutoff for G6PD deficiency could be determined.

- There was substantial variation in the performance and absolute measurements of spectrophotometry conducted in different laboratories, hindering the definition of a universal cutoff for G6PD deficiency.

### What do these findings mean?

- These findings highlight the importance of quality-control measures to minimise the influence of laboratory procedures on observed measurements.

- The data suggest that while a robust universal, assay-specific G6PD activity cutoff value can be established for diagnosis of severe G6PD deficiency ($<30\%$ normal enzyme activity), this approach is less robust for diagnosing intermediate G6PD deficiency.

- Newly developed diagnostic assays that are less sensitive to laboratory conditions and require less sample preparation are required and may help provide more standardised quantitative G6PD activity measurements across different contexts.

## Introduction

*Plasmodium vivax* and *P. ovale* both form dormant liver stages (hypnozoites) that can reactivate weeks to months following an initial infection, resulting in relapsing malaria [1]. The complete treatment of either species requires the use of drugs able to clear hypnozoites from the liver ('radical cure'), alongside standard blood-stage antimalarials. The only licenced antimalarial compounds that can kill hypnozoites and prevent relapses are the 8-aminoquinoline drugs primaquine and tafenoquine. Primaquine is used to treat patients with malaria in two scenarios: either as a single low dose to kill *P. falciparum* gametocytes and reduce transmission, or at a higher dose administered over 14 days to kill *P. vivax* hypnozoites. Tafenoquine, is another 8-aminoquinoline drug, which has recently been licensed as a single-dose hypnozoiticidal agent for *P. vivax* liver stages. While well tolerated in the majority of recipients, standard dosing of either drug can cause severe haemolysis in patients with glucose-6-phosphate dehydrogenase (G6PD) deficiency [2]. Tafenoquine is more slowly eliminated than primaquine; hence, patients are exposed to potentially haemolytic concentrations of the drug for longer. The successful rollout of tafenoquine will therefore require more stringent G6PD screening than is currently available in malaria endemic areas.

G6PD deficiency is a common inherited enzymopathy that is particularly prevalent in malaria-endemic regions [3,4]. Red blood cells (RBCs) of affected individuals are susceptible to haemolysis caused by oxidative stress, induced by a variety of stimuli including drugs (e.g., rasburicase, 8-aminoquinolines, and dapsone), foods (e.g., fava beans), or acute infection [5,6]. The gene encoding G6PD is located on the X chromosome (Xq28). Hence, males inherit a single copy and are either hemizygous G6PD deficient (G6PDd) or G6PD normal (G6PDn), whereas females carry two copies and can be homozygous deficient or normal, or heterozygous for a mutant *G6PD* allele. Hemizygous and homozygous deficient individuals express $>95\%$ deficient RBCs, while heterozygotes harbour two distinct RBC populations, a G6PDd and a G6PDn population. Different patterns of X-inactivation in heterozygotes lead to different proportions of deficient RBCs between individuals and widely varying G6PD phenotypes [7]. Thus, while males are phenotypically either G6PDd or G6PDn, females exhibit a wide range of G6PD levels from very low deficient levels through intermediate to normal. Furthermore, the G6PD gene itself exhibits substantial genetic variability, with over 200 distinct G6PD mutations described [8,9], exhibiting a wide range of enzyme activity levels [10,11].

Ultraviolet (UV) spectrophotometry is the reference standard diagnostic method for quantifying G6PD enzyme activity [12]. Multiple commercial assay kits are available, and although there is strong correlation of measurements between assays, absolute values vary [13,14]. Such absolute values will be important for the universal use of novel point-of-care quantitative diagnostics, such as forthcoming biosensors. The interpretation of quantitative G6PD measurements requires a predetermined definition of 'normal' (100%) G6PD activity. However, there

is no consensus definition of normal activity. Each laboratory must establish its own reference values and diagnostic thresholds [15]. In 2013, Domingo and colleagues proposed a method of standardising this process [16], deriving normal enzyme activity from the median value of non-deficient males, known as the adjusted male median (AMM). Although this approach is less vulnerable to outliers or the underlying prevalence of G6PDd than the standard mean or median, interpretation of assay results requires derivation of a context-specific local AMM. Once defined, the AMM enables classification of samples based on their relative G6PD activity.

Treatment decisions for primaquine or tafenoquine are currently based on two important thresholds: 30% and 70% normal enzyme activity, respectively. The former is the approximate cutoff activity for most qualitative tests [17]. The latter is designed to exclude heterozygous females with intermediate enzyme activity who are also at risk of haemolysis [18,19]. Setting the threshold too low risks falsely categorising patients as G6PDn and exposing individuals to primaquine-induced haemolysis. Setting the threshold too high potentially excludes G6PDn patients from receiving radical cure, putting them at risk of repeated episodes of vivax malaria and associated morbidity.

The aim of the current study was to pool spectrophotometric data from diverse laboratory and geographic contexts to quantify the degree to which assay-based variability influences inter- and intra-study comparability of G6PD spectrophotometry, and to explore the implications of this variability on diagnosing severe and intermediate G6PD deficiency.

## Methods

### Data collation

The data for this prospectively planned meta-analysis were pooled from a systematic literature review and unpublished contributions from collaborating investigators (PROSPERO: CRD42019121414) according to PRISMA guidelines (**S1 File**). Studies involving individual-level spectrophotometric measurements of G6PD activity were identified via a PubMed search (25 April 2018) using the following terms: *G6PD OR "glucose-6-phosphate dehydrogenase" OR "glucose 6 phosphate dehydrogenase") AND (quantitative OR spectrophot*)*. Additional studies were identified via reference lists and correspondence with authors. In view of the wide range of assay manufacturers and laboratory methodologies in the published literature, only papers published between January 2005 and April 2018 were screened for inclusion to ensure comparability of diagnostic protocols and quality control procedures. The title, abstract, and full text of studies between these dates were then screened to identify those that used UV spectrophotometry to define G6PD activity. Only studies that measured NADPH formation at a wavelength of 340 nm were included. Authors were contacted via email and invited to contribute individual patient and quality control data. A minimum of two attempts was made to contact authors before excluding studies. Studies targeting specific ethnic groups (e.g., African-American blood donors) were included, but those that only included individuals with known haematological conditions, family studies, or studies for which insufficient details were available were excluded. Studies of patients with malaria were included but analysed separately from individuals without malaria. One paper included studies performed in two different countries and was considered as two separate data sources ([20]; **S1 Table**).

Demographic data and available haematological parameters were collated and stored in a standardised database along with metadata of the study characteristics. Samples missing data on either G6PD activity (U/g Hb) or sex were excluded from the analysis, as were those for which measured G6PD activity was extreme (>30 U/g Hb). Quality of included studies was assessed using an adapted form of the QUADAS-2 tool ([21]; **S2 File**).

## Data analysis

Intra-laboratory assay repeatability was investigated from replicate measures. For each sample, the coefficient of variation (CV) and absolute difference between replicate measures were calculated, along with a mean CV and mean difference for each study. CV values and proportion of samples with high inter-replicate variability were compared between studies using the Kruskal-Wallis and chi-squared tests.

To assess inter- and intra-laboratory variability in G6PD spectrophotometry, data from the measurement of manufacturer-provided quality control samples were used to quantify the magnitude and variance of control measurements (deficient, intermediate, and normal) and how these differed between assays and studies.

World Health Organization (WHO) prequalification for in vitro diagnostics recently defined G6PD deficiency for males and females at below 30% and intermediate activity between 30% and 80% G6PD activity [22]. The analysis performed here uses the 70% threshold used for the tafenoquine clinical trials [18,19]. For population studies, the AMM [16] was calculated for each study separately and defined as 100% G6PD activity. G6PD deficiency was defined as 'activity below 30% of the respective study AMM' and intermediate G6PD deficiency as 'activity between 30% and 70% of the respective study AMM'. Variation between different study populations was assessed by assay and whether the participant had malaria. Studies with a high risk of bias due to patient selection were excluded from the analysis (**S2 Table**). To preclude the influence of heterozygosity on the observed variability, only male samples were compared. G6PD activity distributions from control and participant samples were compared using the Kruskal-Wallis test with pairwise Mann-Whitney-Wilcoxon tests using Bonferroni correction (magnitude) and Levene's test (variance).

A universal AMM was calculated by applying the standard AMM formula to a comparable subset of all included samples. To control for differences due to assay variability or malaria status, the universal AMM was only derived from samples tested using the Trinity assay in patients without malaria. For each study, separately and across the pooled subset of data, the diagnostic performance (sensitivity, specificity, and negative predictive value [NPV]) of this universal AMM was then evaluated at both the 30% and 70% thresholds, considering diagnoses derived from study-specific AMMs as the reference. Exact binomial confidence limits were estimated for both sensitivity and specificity. Performance was compared to a conservative universal threshold, where 100% G6PD activity was determined by the upper limit of the 95% CI of the mean G6PD activity for G6PDn males in the included subset. Due to the limited availability of relevant covariate data from included studies, the universal AMM, conservative threshold, and threshold performances were not adjusted for study-level covariates. Furthermore, such adjustment for study-level covariates would impose an impractical level of complexity in clinical practice.

All data management, visualisation, and statistical analyses were performed using the tidyverse [23], ggplot2 [24], cowplot [25], epiR [26], and base R packages in R version 3.5.2 [27]. Prior ethical approval was obtained for all primary studies included in the analysis.

## Results

In total, 312 studies were identified from the literature published between January 2005 and April 2018, as well as 18 unpublished studies. Of these, 243 studies were excluded based on title, abstract, or full text review, and 55 studies were excluded because the corresponding author did not respond or the relevant data were not available (**Fig 1**). A further two studies were excluded due to methodological criteria (incomparable measurement units or spectrophotometry wavelength). Data from 231 individuals were excluded for reasons stated in **Fig 1**.

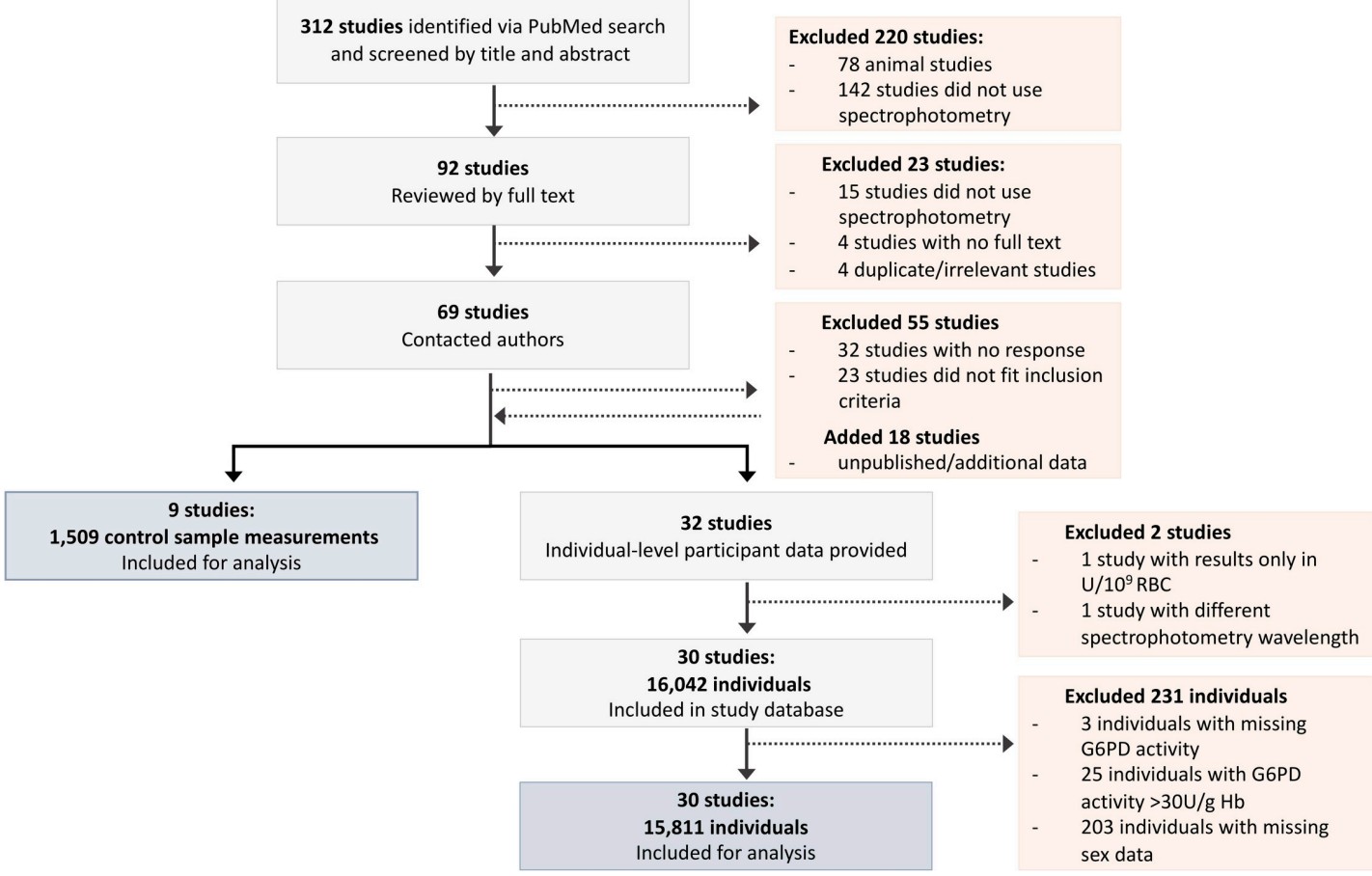

**Fig 1. PRISMA flow diagram depicting the literature search and data-screening procedures.** G6PD, glucose-6-phosphate dehydrogenase; RBC, red blood cell.

The final dataset comprised spectrophotometric measurements from 15,811 individuals collected from 30 studies (20 in Asia, 5 in Africa, and 5 in the Americas; **Table 1**). The age of participants ranged from 0 to 86 years, and 44.4% of participants (7,083 individuals) were female. G6PD spectrophotometry results were generated from six different manufacturer's assays: Trinity Biotech, Ireland (21 studies, 12,222 participants); Randox Laboratories, United Kingdom (4 studies, 1,476 participants); Pointe Scientific, United States (2 studies, 883 participants); Sigma-Aldrich, US (2 studies, 691 participants); BIOLABO, France (1 study, 320 participants), and Spinreact, Spain (1 study, 319 participants).

## Assay repeatability

Replicate measurements of G6PD activity were available from five studies, including 14.4% (2,204/15,811) of samples (**Table 2** and **Fig 2**). One study ($n = 609$) performed triplicate measures on all samples, while four studies ($n = 1,595$) performed duplicate measurements on all samples and a third test on a subset of samples with high inter-replicate variability ($n = 53$). The magnitude of inter-replicate variability differed between studies, with the study-specific maximum difference ranging from 0.65 U/g Hb to 8.64 U/g Hb and the mean CV ranging from 1.6% to 14.9% (**Table 2**). The percentage of samples that showed a high inter-replicate difference (>2 U/g Hb) differed significantly across all studies ($p < 0.001$), ranging from 0% to 32% of each study's samples. Similar inter-study differences were evident when considering

**Table 1. Characteristics of the final pooled database by malaria status.**

| Characteristics | Malaria Positive | | Malaria Negative | | Malaria Unknown | | Total | |
|---|---|---|---|---|---|---|---|---|
| **Sex** | | | | | | | | |
| Male | 3,433 | (21.71) | 3,360 | (21.25) | 1,990 | (12.59) | **8,783** | **(55.55)** |
| Female | 3,446 | (21.79) | 1,422 | (8.99) | 2,160 | (13.66) | **7,028** | **(44.45)** |
| **Region** | | | | | | | | |
| Africa | 808 | (5.11) | 708 | (4.48) | 127 | (0.8) | **1,643** | **(10.39)** |
| Americas | 1,588 | (10.04) | 324 | (2.05) | 346 | (2.19) | **2,258** | **(14.28)** |
| Southeast Asia | 4,483 | (28.35) | 3,750 | (23.72) | 3,677 | (23.26) | **11,910** | **(75.33)** |
| **Age** | | | | | | | | |
| 0 to 1 | 115 | (0.73) | 8 | (0.05) | 566 | (3.58) | **689** | **(4.36)** |
| 1 to 5 | 601 | (3.8) | 194 | (1.23) | 123 | (0.78) | **918** | **(5.81)** |
| 5 to 15 | 1,250 | (7.91) | 930 | (5.88) | 1,104 | (6.98) | **3,284** | **(20.77)** |
| 15 to 60 | 3,408 | (21.55) | 2,939 | (18.59) | 2,099 | (13.28) | **8,446** | **(53.42)** |
| >60 | 179 | (1.13) | 29 | (0.18) | 50 | (0.32) | **258** | **(1.63)** |
| Unknown | 1,326 | (8.39) | 682 | (4.31) | 208 | (1.32) | **2,216** | **(14.02)** |
| **G6PD Assay***  | | | | | | | | |
| Trinity | 5,119 | (32.17) | 3,964 | (24.91) | 3,139 | (19.73) | **12,222** | **(77.3)** |
| Randox | 1,296 | (8.15) | 180 | (1.13) | - | - | **1,476** | **(9.34)** |
| Pointe Scientific | 564 | (3.54) | 319 | (2) | - | - | **883** | **(5.58)** |
| Sigma-Aldrich | - | - | - | - | 691 | (4.34) | **691** | **(4.37)** |
| BIOLABO | - | - | - | - | 320 | (2.01) | **320** | **(2.02)** |
| Spinreact | - | - | 319 | (2) | - | - | **319** | **(2.02)** |
| **Total** | **6,879** | **(43.51)** | **4,782** | **(30.24)** | **4,150** | **(26.25)** | **15,811** | **(100)** |

Values indicate the number of individuals and percentage of the total database, *n* (%), in each cell.

*One hundred individuals tested by both Trinity and Randox are counted twice here.

relative variability, with the percentage of samples exhibiting a high CV (>10%) ranging from 1.53% to 62%.

## Inter- and intra-laboratory variability

Quality control data were available from nine studies: seven that used the Trinity assay (using normal, intermediate, and deficient controls) and three studies that used the Randox assay

**Table 2. Inter-replicate differences in G6PD spectrophotometry.**

| Study | Assay | Sample Size | Mean Δ[a] (Range) | Δ >2 U/g Hb (% Samples) | Mean CV (Range) | CV >10% (% Samples) |
|---|---|---|---|---|---|---|
| **Bancone et al., 2015 [28]** | Trinity | 150 | 0.1 (0.0–0.7) | 0.0 | 2.5 (0.0–23.2) | 3.3 |
| **Hailu et al., 2018[b]** | Trinity | 367 | 0.7 (0.0–2.7) | 1.6 | 4/0 (0.0–15.7) | 4.9 |
| **Ley et al., 2018[b]*** | Randox | 92 | 0.6 (0.0–1.9) | 0.0 | 9.8 (0.0–48.8) | 34.8 |
| **Ley et al., 2018[b]*** | Trinity | 100 | 1.8 (0.0–8.6) | 32.0 | 14.9 (0.3–60.5) | 62.0 |
| **Oo et al., 2016 [29]** | Trinity | 978 | 0.2 (0.0–3.6) | 0.5 | 1.6 (0.0–35.2) | 1.5 |
| **Satyagraha et al., 2016 [30]** | Trinity | 609 | 1.0 (0.0–7.8) | 12.6 | 6.7 (0.0–140) | 16.3 |

[a]Δ = absolute inter-replicate difference (U/g Hb).

[b]Unpublished study.

*Note: These rows refer to a single study which provided measurements of the same samples using both Randox and Trinity.

Abbreviation: CV, coefficient of variation (%)

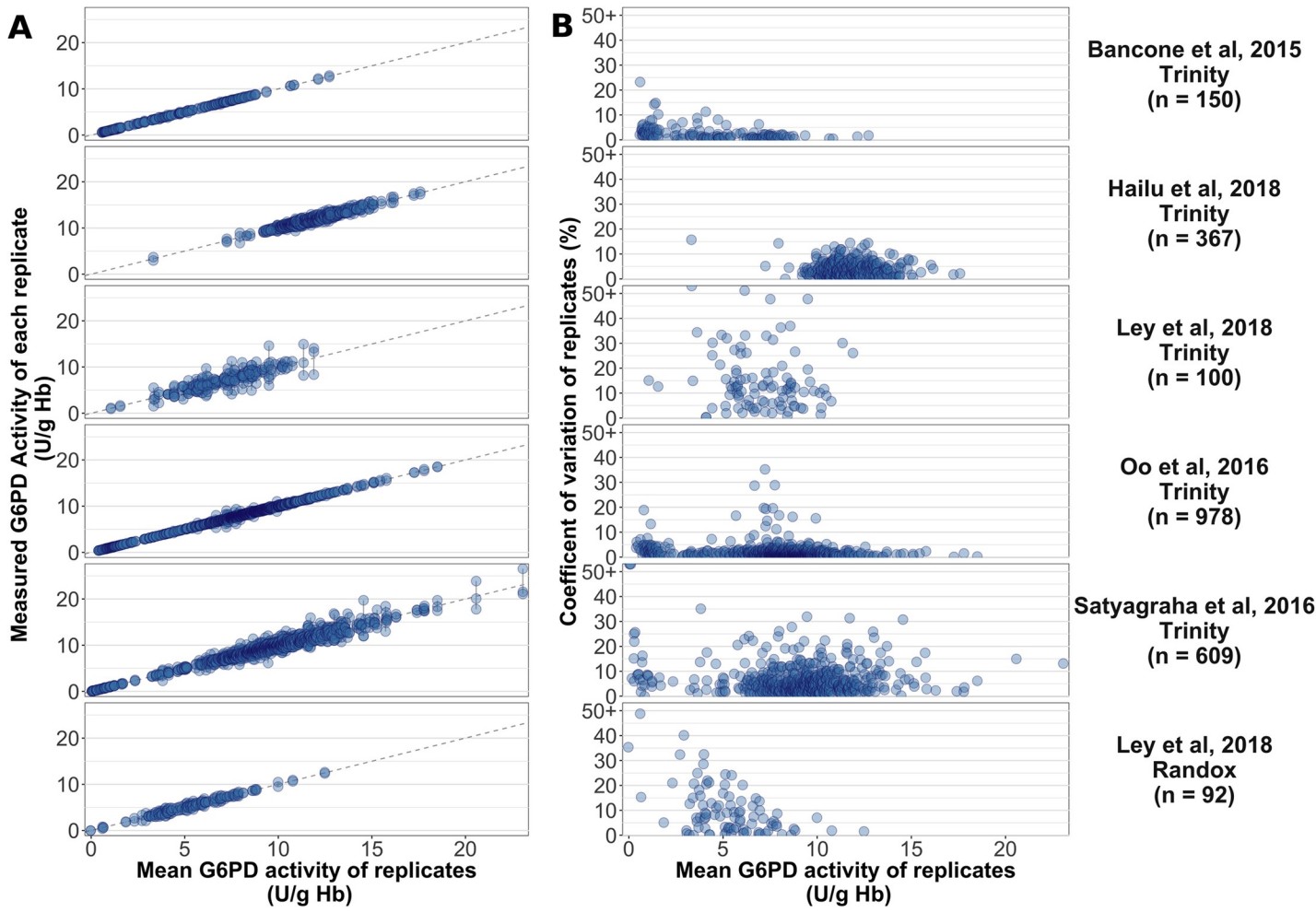

**Fig 2. Repeatability of G6PD spectrophotometry.** Inter-replicate agreement in five studies enrolling 2,204 participants. **(A)** Absolute G6PD activity (U/g Hb) of replicate measures. Each point represents a single G6PD activity measurement, with measurements from the same individual connected by vertical lines for clarity. **(B)** Relative difference (CV; %) between replicate measures of G6PD activity. Five points with CV greater than 50% are shown as '50+' for clarity (CV = 51, 61, 98, 110, 140). For both panels, the assay used is labelled for each study and the study-wise mean (range) inter-replicate difference is shown. Note: One study, Ley et al., 2018, provided measurements from the same samples using both Randox ($n$ = 92) and Trinity ($n$ = 100). CV, coefficient of variation; G6PD, glucose-6-phosphate dehydrogenase.

(normal and deficient controls; **Fig 3**). Results differed significantly across studies for all control categories ($p < 0.001$), when considering both the Trinity and Randox assays. Smaller inter-study differences were observed between the studies using Randox, all of which were conducted in the same laboratory. Intra-study variability (study-level variance of control measurements) differed significantly for all control categories for studies using both the Trinity (normal, $p < 0.001$; intermediate, $p < 0.001$; deficient, $p < 0.001$), and Randox (normal, $p = 0.002$; deficient, $p = 0.02$) assays.

To control for differences attributable to assay method, the analysis of the variability in spectrophotometry data between study populations was first addressed in the 18 studies (6,245 individuals) using the Trinity assay. There was considerable inter-study variation in the distribution of G6PD activity and the derived AMM values across studies (**Fig 4**). The AMM ranged from 5.7 to 12.6 across studies consisting of participants without malaria and 7.8 U/g Hb to 12.4 U/g Hb for those with malaria. The inter-study differences in G6PD activity were

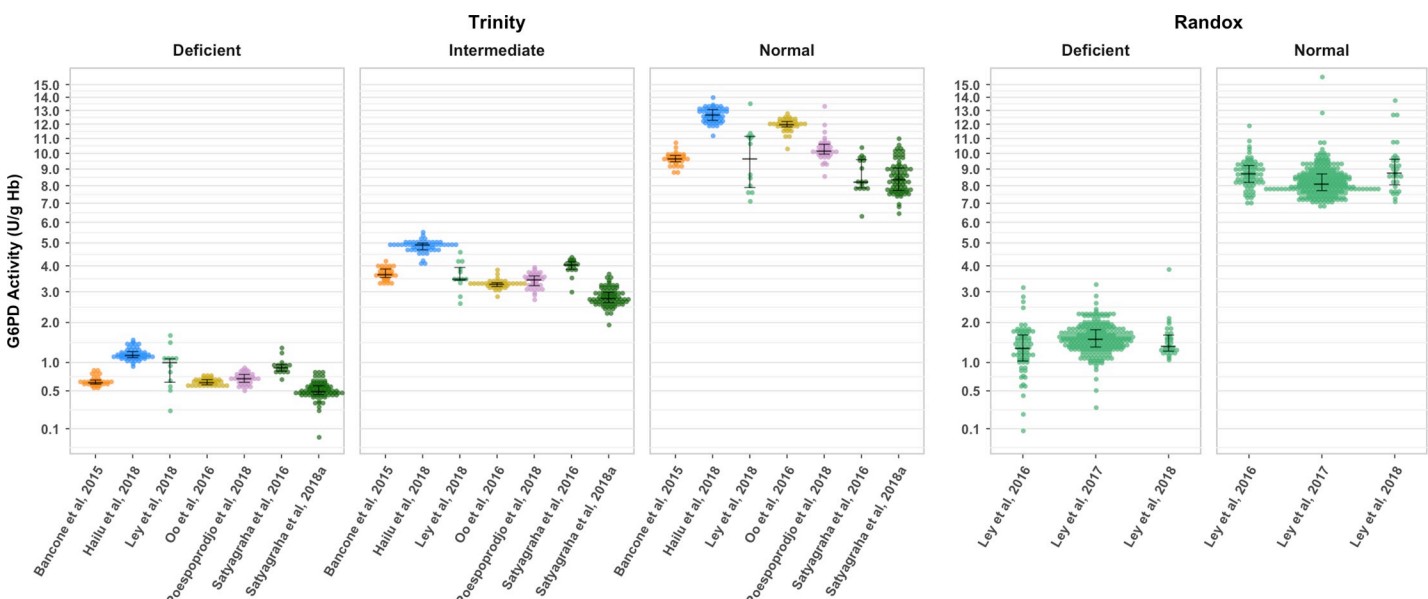

**Fig 3. Inter-study variation in control sample measurements.** Quality control data from nine studies are shown ($n = 1,509$). The three leftmost panels contain measurements of deficient, intermediate, and normal controls from studies using the Trinity assay; panels on the right depict deficient and normal control measurements using the Randox assay. The median and interquartile range for each category are superimposed in black. Points are coloured to indicate studies conducted in the same laboratory. A square-root transformation is applied to the y-axis to depict variation in deficient samples more clearly. Note: One study, Ley et al., 2018, provided measurements from the same samples using both Randox ($n = 92$) and Trinity ($n = 100$). G6PD, glucose-6-phosphate dehydrogenase.

statistically significant across all studies ($p < 0.05$). In the studies that used a spectrophotometry assay other than Trinity, there was significant inter-study variation in G6PDn males for Pointe Scientific ($p < 0.001$), Randox ($p = 0.04$), and Sigma-Aldrich ($p < 0.001$) (**S1 Fig**).

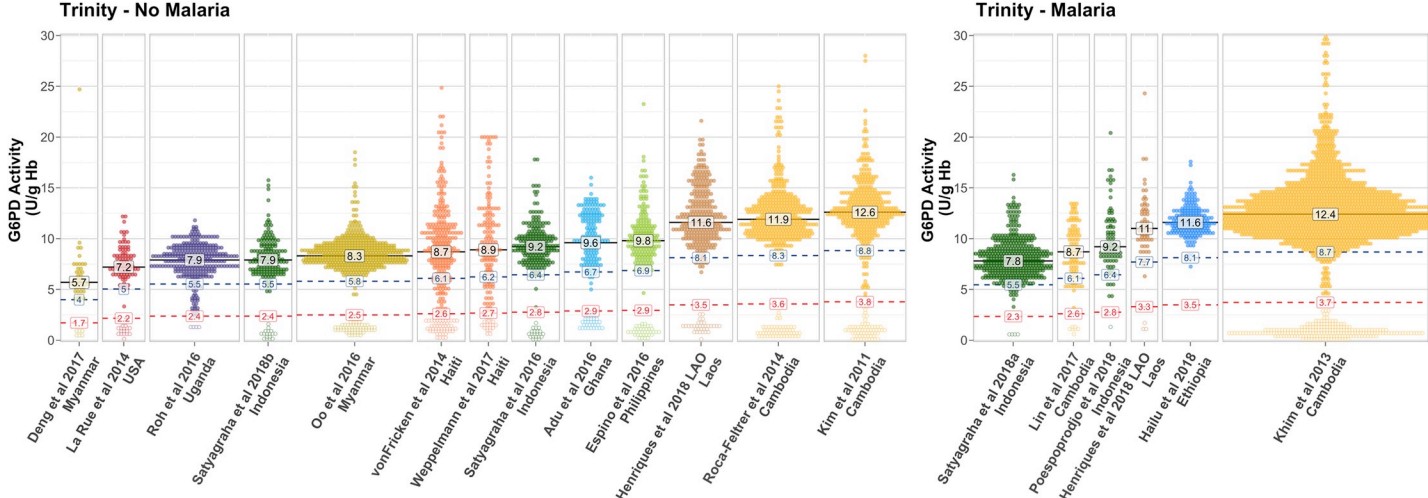

**Fig 4. Inter-study variation in G6PD activity measured using the Trinity assay.** The distribution of G6PD activity measurements is shown for males enrolled in representative studies using the Trinity G6PD assay. Participants with ($n = 2,694$) and without malaria ($n = 3,516$) are shown separately. The AMM (black text), 70% threshold (blue), and 30% threshold (red) are labelled for each study (U/g Hb). Point colours indicate the country of origin of each study. Filled points represent G6PDn individuals (>30% study AMM) and hollow points indicate G6PDd individuals (<30% study AMM). Note: Roh et al., 2016, included 63 males <1 year of age, who may have had elevated G6PD activity [31,32]. AMM, adjusted male median; G6PD, glucose-6-phosphate dehydrogenase; G6PDd, G6PD deficient; G6PDn, G6PD normal.

**Table 3. Performance of pooled universal AMM diagnostic thresholds.**

| | Deficient Threshold (30%) | | |
|---|---|---|---|
| | Sensitivity | Specificity | NPV |
| **Males** | 0.99 (0.97–1.00) | 1.00 (1.00–1.00) | 1.00 (1.00–1.00) |
| **Females** | 0.91 (0.85–0.95) | 1.00 (0.99–1.00) | 1.00 (0.99–1.00) |
| **Overall** | 0.97 (0.95–0.98) | 1.00 (1.00–1.00) | 1.00 (1.00–1.00) |
| | Intermediate Threshold (70%) | | |
| | Sensitivity | Specificity | NPV |
| **Males** | 0.95 (0.93–0.96) | 0.96 (0.95–0.96) | 0.99 (0.98–0.99) |
| **Females** | 0.84 (0.81–0.87) | 0.96 (0.95–0.97) | 0.97 (0.96–0.97) |
| **Overall** | 0.89 (0.87–0.91) | 0.96 (0.95–0.96) | 0.98 (0.97–0.98) |

Abbreviations: AMM, adjusted male median; NPV, negative predictive value

## Universal thresholds

Participants were categorised as being either severely deficient (<30% AMM; ineligible for primaquine) or severely/intermediately deficient (<70% AMM; ineligible for tafenoquine) using the study-specific AMMs. In patients without malaria assessed using the Trinity assay, the pooled AMM (i.e., 100% G6PD activity threshold) was 9.4 U/g Hb, resulting in a 70% threshold of 6.6 U/g Hb and a 30% threshold of 2.8 U/g Hb. The conservative universal threshold (calculated as the upper limit of the 95% CI of the mean G6PD activity, instead of the median used for the AMM) yielded a 100% threshold of 9.7 U/g Hb; 70% threshold of 6.8 U/g Hb, and 30% threshold of 2.9 U/g Hb.

Using the site-specific AMM as the reference, the pooled AMM correctly categorised 89% to 100% of severely deficient patients (both males and females) at the 30% threshold with a pooled sensitivity of 97% (95% CI: 95%–98%). At this threshold, the study-wise specificity was greater than 96% for all studies with a pooled specificity of 100% (95% CI: 100%–100%) (**Table 3**, **S3 File**: **S7–S12 Figs**). Seventeen individuals (12 females and 5 males, out of a total of 7,520) were falsely classified as G6PDn, with a mean (range) G6PD activity of 26.8% (23%–29.6%) of the local AMM (**S4 Table**, **Fig 5**, **S3 File**: **S2 and S3 Figs**).

At the 70% threshold, the study-wise sensitivity of the universal AMM ranged between 64% and 100% with a pooled sensitivity of 89% (95% CI: 87%–91%). At this threshold, the specificity ranged from 35% to 100%, with a pooled specificity of 96% (95% CI: 95%–96%) (**Table 3** & **S3 File**: **S7–S12 Figs**). One hundred and thirty-nine individuals, 107 females and 32 males, were falsely classified as G6PDn at the 70% level, with a mean (range) G6PD activity of 56.5% (52.4%–59.5%) of the local AMM (**S4 Table**, **Fig 5**, **S3 File**: **S2 and S3 Figs**). At both the 30% and 70% threshold, performance was slightly improved when considering the conservative universal thresholds (**S3 and S4 Tables**, **S3 File**: **S4–S6, S13–S19 Figs**). Diagnostic performance of universal thresholds did not differ substantially in patients with or without malaria. At all thresholds, when there was a difference in diagnostic performance of universal thresholds, the performance in females was worse than in males, across all studies and within individual studies (**Table 3 and S3 File**: **S8–S13 Figs**).

## Discussion

Safe implementation of radical cure of malaria with primaquine or tafenoquine will be critical for the timely elimination of *P. vivax*. In view of the risk of drug-induced haemolysis, patients should be tested for G6PD deficiency prior to treatment to avoid exposing vulnerable individuals to

**Fig 5. Universal diagnostic classifications by G6PD activity—both males and females, using pooled AMM.** Relative G6PD activity is shown (x-axis, normalised to site-specific AMM) for individuals falling into each diagnostic category, as depicted by coloured bars (false normal [FN], red; false deficient [FD], orange; true normal [TN], light grey; true deficient [TD], black). Diagnoses are shown at both the 30% threshold (left; 17 FN, 21 FD, 6,926 TN, 556 TD) and 70% threshold (right; 139 FN, 258 FD, 5,983 TN, 1,140 TD). All individuals tested with Trinity without malaria infection are shown, except 69 individuals with G6PD activity >200% local AMM (*n* = 7,451). AMM, adjusted male median; G6PD, glucose-6-phosphate dehydrogenase.

oxidative stress. Widespread application of radical cure will thus require robust and easily interpretable diagnostic thresholds for G6PD enzyme activity. Our pooled analysis of G6PD activity across a diverse range of studies and geographical locations highlights significant inter-study and intra-study differences in absolute G6PD measurements derived using spectrophotometry. This variability has strong potential to confound reliable diagnosis of G6PD deficiency.

We observed considerable variability in G6PD activity measurements between sites, even when considering the same spectrophotometric assay. The differences in the quantification of control samples suggest substantial contribution of laboratory- or assay-based factors, although these may be exacerbated by unmeasured genetic or environmental differences between studies. Despite the large sample size of this meta-analysis, the data lacked the granular information necessary to isolate specific laboratory or procedural factors at play. The presence of differences between research laboratories illustrates the likely pervasiveness of interlaboratory differences in G6PD spectrophotometry, with fundamental implications for comparing absolute G6PD activity measurements between studies.

Despite this variability, our findings demonstrate strong performance of a universal threshold for identifying G6PD activity below 30% of the local AMM. At this level, the universal diagnostic threshold (2.8 U/g Hb) demonstrated robust diagnostic performance with sensitivity and specificity exceeding 97%. In these cohorts, 17 out of 7,520 individuals would receive primaquine despite having a G6PD activity less than 30% the local AMM; however, all of these misdiagnoses occurred around the diagnostic cutoff, with a minimum G6PD activity of 23% (S4 Table, Fig 5). The majority of the 17 misdiagnoses were in females, suggesting that a portion of these may come from heterozygous females with G6PD activities spanning the 30% threshold. Hence, this threshold may have utility in certain contexts where a local AMM is unavailable (e.g., validation studies of qualitative assays).

The use of tafenoquine requires a more stringent threshold to reduce the risk of haemolysis in heterozygous females with intermediate deficiency. At the 70% enzyme activity level, the

diagnostic performance of pooled universal cutoffs was worse than that for the 30% threshold. This is likely a consequence of both inter-study variation in G6PD activity, as well as natural variation (noise) in G6PD activity levels around this 70% limit. Of the 139 individuals misclassified as G6PDn at this level, only 36 (25.4%) exhibited a G6PD activity less than 60% of the local AMM, and all had a G6PD activity greater than 53%. Again, similar to the 30% threshold, the majority of these were females that are in the 60%–70% activity range. The exact relationship between G6PD activity and the haemolytic risk associated with tafenoquine is unknown, although the risk is thought to be inversely correlated with enzyme activity [33]. Reassuringly, false normal diagnoses at the 70% enzyme activity level using the universal thresholds occurred in a minority of individuals with G6PD activity mostly near the 70% mark.

The universal thresholds defined in our study are based on the Trinity assay kit, which is now discontinued. While Alam and colleagues have shown good correlation between Trinity measurements and results from a Pointe Scientific assay, with little difference in absolute measurements ($n$ = 50, [14]), a study by Pal and colleagues found a more modest correlation in absolute measurements ($n$ = 183; [13]). Consequently, demonstration of suitability of a proposed universal threshold will need to be demonstrated with each specific assay prior to its widespread endorsement and application. Promising point-of-care quantitative biosensors reduce the need for sample storage and complex processing [12,34]. As such, these new assays may exhibit superior interlaboratory comparability and in future may provide the basis for a universal definition of G6PD deficiency. Until there is consensus regarding safe and robust universal thresholds, site- and assay-specific definitions of G6PD deficiency will still be required. Currently, laboratories either use the product insert ranges or determine normal ranges based on normal samples tested in the laboratory.

We identified notable differences in assay repeatability and variance of control measurements between sites. Such assay-based variability may arise from inconsistencies in assay procedures, or sample handling [16,34,35]. This variation demonstrates the importance of performing and monitoring replicate measurements and control sample measurements in order to minimise assay-based variability and maximise comparability of results.

It is worth noting that diagnostic definitions of 'intermediate' G6PD activity have varied over time and contexts. The early WHO classifications of G6PD variants considered anything above 60% to be normal G6PD activity [36], current WHO guidelines place the cutoff at 80% normal activity [17], and prescription requirements for tafenoquine in the US and Australia require a G6PD activity of 70% or higher [37]. Regardless of the definition, our study highlights the significant challenges in establishing an estimate of 100% G6PD activity (in clinically relevant units of U/g Hb) from which patients' relative enzyme activity can be calculated.

Our study has a number of limitations. First, although we excluded studies in which haematological conditions may have influenced observed G6PD activity levels, the final dataset consisted of samples from diverse clinical and community survey contexts. This may have led to unequal influence of undiagnosed health conditions (e.g., other haematological conditions) upon measured G6PD activity. However, there was no clear pattern in G6PD activity by study type. Furthermore, while it has been suggested that neonates have elevated G6PD activity [31,32], introducing possible bias, few of the included studies enrolled newborns ([38,39]). Second, it is common practice to exclude replicate measures that differ by more than a certain amount (e.g., CV > 10%) as well as control measurements falling outside of an expected range. Included datasets may have already excluded these values in some, but not all, cases, meaning that the current study would overestimate true assay performance. Third, the study did not consider lot-to-lot variability in control isolates (which consist of lyophilised blood specimens) and assay reagents; these may have contributed to some of the inter-study variability observed

for both the controls and AMM values. Nevertheless, the current findings represent an indication of inter- and intra-study variability in 'valid' results of G6PD spectrophotometry.

## Conclusions

Interlaboratory variability hinders the definition of universal cutoff values for the classification of G6PD activity using spectrophotometry, particularly at the 70% G6PD activity level. Caution is advised in comparing research findings based on absolute G6PD activity measurements across studies, such as those characterising novel variants or assessing clinical safety in patients exposed to 8-aminoquinolines. In these cases, the derivation of relative G6PD activity using the AMM remains a more appropriate approach. Because assay precision varies considerably between laboratories, the use of replicate measures and control sample measurements is crucial to ensure quality control. Clinical laboratories typically provide the patients' G6PD value and a normal G6PD range for the laboratory, which is often determined such that it discriminates at or above the 70% threshold required to prescribe tafenoquine. Novel point-of-care assays, such as recently developed quantitative biosensors, are currently being evaluated in field trials. These assays are designed to require less sample preparation and offer more robust performance across diverse temperature ranges and clinical contexts than spectrophotometry. As such, they may provide superior inter-site comparability than the current reference standard of spectrophotometry; however, until this has been shown, routine quantitative diagnosis of G6PDd will require site- and assay-specific local definitions of G6PD activity to ensure that tafenoquine can be administered safely. In any case, no diagnostic assay is perfect, meaning that radical cure treatment policies must be accompanied by patient and health worker training on the warning signs and risks of haemolysis, along with access to transfusion services when needed.

## Supporting information

**S1 File. PRISMA checklist.**
(PDF)

**S2 File. Adapted QUADAS-2 [21] tool used for quality appraisal.** QUADAS-2, Quality Assessment of Diagnostic Accuracy Studies-2.
(PDF)

**S3 File. Additional figures indicating diagnostic performance for alternative universal thresholds.**
(PDF)

**S4 File. List of data contacts for each included study.**
(PDF)

**S1 Fig. Inter-study variability for assays other than Trinity.** Comparison of G6PD activity distributions among males. **(A, B)** Studies using an assay other than Trinity, by malaria status, and **(C)** studies with purposive sampling. Assays used are indicated in brackets on the x-axis. The AMM (black text), 30% threshold (red), and 70% threshold (blue) are labelled for each study (in U/g Hb). Point colours indicate the country of origin of each study. Filled points represent G6PDn individuals (>30% study AMM), and hollow points indicate G6PDd individuals (<30% study AMM). Note: 100% G6PD activity shown for Henriques et al., 2018, and Bancone et al., 2015 **(C)**, are as reported, not the AMM, due to strong oversampling of G6PDd individuals. Charoenkwan et al. [36] enrolled neonates, who may have had elevated G6PD activity [31, 32]. AMM, adjusted male median; G6PD, glucose-6-phosphate dehydrogenase;

G6PDd, G6PD deficient; G6PDn, G6PD normal.
(PNG)

**S1 Table. Characteristics of data sources included in the pooled database.**
(XLSX)

**S2 Table. Quality appraisal of included studies using adapted QUADAS-2 [21] tool.** QUADAS-2, Quality Assessment of Diagnostic Accuracy Studies-2.
(XLSX)

**S3 Table. Performance of a conservative universal diagnostic threshold.**
(XLSX)

**S4 Table. Frequency and G6PD activity of false normal individuals using universal diagnostic thresholds.** G6PD, glucose-6-phosphate dehydrogenase.
(XLSX)

## Acknowledgments

We would like to thank all participants of the included studies, as well as all staff who contributed to the primary included articles and assisted in sharing their datasets.

**Disclaimer:** The views expressed here are solely those of the authors and do not reflect the views, policies or positions of the US Government or Department of Defense. Material has been reviewed by the Walter Reed Army Institute of Research. There is no objection to its presentation and/or publication. The opinions or assertions contained herein are the private views of the author, and are not to be construed as official, or as reflecting true views of the Department of the Army or the Department of Defense. The investigators have adhered to the policies for protection of human subjects as prescribed in AR 70–25.

## Author Contributions

**Conceptualization:** Daniel A. Pfeffer, Benedikt Ley, Rosalind E. Howes, Gonzalo J. Domingo, Ric N. Price.

**Data curation:** Daniel A. Pfeffer, Benedikt Ley, Ari Winasti Satyagraha, Germana Bancone.

**Formal analysis:** Daniel A. Pfeffer, David J. Price.

**Funding acquisition:** Ric N. Price.

**Investigation:** Daniel A. Pfeffer, Benedikt Ley, Patrick Adu, Mohammad Shafiul Alam, Pooja Bansil, Yap Boum, II, Marcelo Brito, Pimlak Charoenkwan, Liwang Cui, Zeshuai Deng, Ochaka Julie Egesie, Fe Esperanza Espino, Michael E. von Fricken, Muzamil Mahdi Abdel Hamid, Yongshu He, Gisela Henriques, Wasif Ali Khan, Nimol Khim, Saorin Kim, Marcus Lacerda, Chanthap Lon, Asrat Hailu Mekuria, Didier Menard, Wuelton Monteiro, François Nosten, Nwe Nwe Oo, Sampa Pal, Duangdao Palasuwan, Sunil Parikh, Ayodhia Pitaloka Pasaribu, Jeanne Rini Poespoprodjo, Arantxa Roca-Feltrer, Michelle E. Roh, David L. Saunders, Michele D. Spring, Inge Sutanto, Kamala Ley-Thriemer, Thomas A. Weppelmann, Lorenz von Seidlein, Ari Winasti Satyagraha, Germana Bancone, Gonzalo J. Domingo, Ric N. Price.

**Methodology:** Daniel A. Pfeffer, Benedikt Ley, Ari Winasti Satyagraha, Germana Bancone.

**Resources:** Ric N. Price.

**Supervision:** Benedikt Ley, Rosalind E. Howes, Archie Clements, Lorenz von Seidlein, Germana Bancone, Gonzalo J. Domingo, Ric N. Price.

**Visualization:** Daniel A. Pfeffer.

**Writing – original draft:** Daniel A. Pfeffer.

**Writing – review & editing:** Daniel A. Pfeffer, Benedikt Ley, Rosalind E. Howes, Patrick Adu, Mohammad Shafiul Alam, Pooja Bansil, Yap Boum, II, Marcelo Brito, Pimlak Charoenkwan, Archie Clements, Liwang Cui, Zeshuai Deng, Ochaka Julie Egesie, Fe Esperanza Espino, Michael E. von Fricken, Muzamil Mahdi Abdel Hamid, Yongshu He, Gisela Henriques, Wasif Ali Khan, Nimol Khim, Saorin Kim, Marcus Lacerda, Chanthap Lon, Asrat Hailu Mekuria, Didier Menard, Wuelton Monteiro, François Nosten, Nwe Nwe Oo, Sampa Pal, Duangdao Palasuwan, Sunil Parikh, Ayodhia Pitaloka Pasaribu, Jeanne Rini Poespoprodjo, David J. Price, Arantxa Roca-Feltrer, Michelle E. Roh, Michele D. Spring, Inge Sutanto, Kamala Ley-Thriemer, Thomas A. Weppelmann, Lorenz von Seidlein, Ari Winasti Satyagraha, Germana Bancone, Gonzalo J. Domingo, Ric N. Price.

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
