## [Decision Letter · Decision Letter 0]

8 Oct 2019

Dear Dr. Pfeffer,

Thank you very much for submitting your manuscript "Quantification of glucose-6-phosphate dehydrogenase activity by spectrophotometry: A systematic review and meta-analysis" (PMEDICINE-D-19-02590) for consideration at PLOS Medicine. 

Your paper was evaluated by a senior editor and discussed among all the editors here. It was also discussed with an academic editor with relevant expertise, and sent to three independent reviewers, including a statistical reviewer. The reviews are appended at the bottom of this email and any accompanying reviewer attachments can be seen via the link below:

[LINK]

In light of these reviews, I am afraid that we will not be able to accept the manuscript for publication in the journal in its current form, but we would like to consider a revised version that addresses the reviewers' and editors' comments. Obviously we cannot make any decision about publication until we have seen the revised manuscript and your response, and we plan to seek re-review by one or more of the reviewers. 

We expect to receive your revised manuscript by Oct 29 2019 11:59PM. Please email us (plosmedicine@plos.org) if you have any questions or concerns.

We look forward to receiving your revised manuscript. 

Sincerely,

Thomas McBride, PhD

Senior Editor 

PLOS Medicine

plosmedicine.org

1- Thank you for providing your PRISMA statement. Please replace the page numbers with paragraph numbers per section (e.g. "Methods, paragraph 1"), since the page numbers of the final published paper may be different from the page numbers in the current manuscript.

2- Thank you for agreeing to include the aggregate data in the supplementary files. However, I did not see a data file, please include this with your revision. Additionally, it would be helpful to include a supplementary text file listing the data contacts for each of the studies included in your analysis.

3- In the Abstract Methods and findings, please provide the dates of search, data sources, types of study designs included, eligibility criteria, and synthesis/appraisal methods. 

4- In the last sentence of the Abstract Methods and Findings section, please describe the main limitation(s) of the study's methodology.

5- Thank you for including an Author Summary. Please reformat the Author Summary into 2-3 single sentence bullet points for each of the following questions. Bullet points should be objective, brief, succinct, specific, accurate, and avoid technical language.

>Why Was This Study Done? Authors should reflect on what was known about the topic before the research was published and why the research was needed.

>What Did the Researchers Do and Find? Authors should briefly describe the study design that was used and the study’s major findings. Do include the headline numbers from the study, such as the sample size and key findings. 

>What Do These Findings Mean? Authors should reflect on the new knowledge generated by the research and the implications for practice, research, policy, or public health. Authors should also consider how the interpretation of the study’s findings may be affected by the study limitations.

Please see our author guidelines for more information: https://journals.plos.org/plosmedicine/s/revising-your-manuscript#loc-author-summary

Comments from the reviewers:

Reviewer #2: Pfeffer and colleagues have presented the results of a systematic review and meta-analyses of individual participant data evaluating lab-based variation in diagnostic results for spectrophotometry derived diagnosis for G6PD deficiency malaria and non-malaria samples. They further pool data to derive a universal adjusted male median (AMM) from a sub-set of samples derived from Trinity assays and apply this threshold to define G6PD deficiency compared to lab-specific thresholds as the reference standard. This resulted in identifying deficiency with 89-100% sensitivity and 96-100% specificity. I commend the authors for a large undertaking in collating the international study data globally and the results could potentially have some policy implications on determining a universal reference standard. There are a few comments I have on the methodology, in particular relating to how the samples were pooled that should be clarified and addressed prior to publication. 

The main issue the authors should clarify is that it isn't clear how the authors pooled the AMM in the IPD and how they accounted for between study heterogeneity (starting 257). The authors are rather vague in their presentation of the statistical methods here: "A pooled universal AMM was calculated across a comparable subset of all included samples." Given the challenges in pooling medians across studies (non-normality assumed), I would presume the authors would have had to calculate a AMM across the entirety of the pooled sample directly from the pooled data in one-stage, and hence one of the limitations may in fact be not being able to account for between study-heterogeneity (as the dominant IPD methodology if pooling means would be to account for this in an error-term in the regression model). This would be an important consideration/limitation as that 75% of the samples come from Asia (Table 1) and study labs have large variations: the authors report range of difference from 0.65- 8.64 with coefficient of variation between 1.6% to 14.9%. Inter-replicate difference was significantly difference between studies. Despite showing that 5/6 assays were from Trinity assay - this still had high levels of variability. 

This means that in Line 263 where a conservative threshold was derived as the upper bound of the 95% CI of the mean G6PD activity, the pooled mean G6DP activity should in fact take into consideration between study differences. As this can be derived from a pooled regression analyses, covariate adjustments, study heterogeneity can be incorporated into a hierarchical model either in one-stage of two-stages. It's unclear how the authors pooled the mean here as well and it would beneficial to clarify the methods. 

Finally, it's nice to see the authors give an overall pooled sensitivity and specificity (Lines 365-367), but it's also not described in the methods how this was done and if study-level covariates were incorporated in these pooled sensitivity and specificity. For completeness of presentation, it would be also useful to present PPV, NPV at varying prevalence and derive an overall accuracy measure summary ROC plot. See Riley et al. 2008 (https://onlinelibrary.wiley.com/doi/epdf/10.1002/sim.3441) and Cochrane handbook suggests having this metric: https://methods.cochrane.org/sites/methods.cochrane.org.sdt/files/public/uploads/Chapter%2010%20-%20Version%201.0.pdf

Other items to address: 

Line 206: It's still not clear why the lower bound for was 2005 for laboratory changes and methodology. Was there some particular change around this time or was this just a pragmatic choice to limit the number of studies? 

Line starting 243: The analysis uses AMM to define 100% G6PD activity, then uses WHO definitions to define categories for < 30% of the AMM is deficiency and activity between 30-70% for intermediate from tafenoquine clinical trials and not the recent WHO definition between 30-80% activity. It would be advantageous to benchmark also against the recent WHO definition as this should enhance the results policy implications (30-80% definition for intermediate activity)

Despite these suggestions above - this is a comprehensive review and should warrant consideration for publication. 

Reviewer #3: This is an important and timely paper as countries with Vivax malaria especially those aiming to eliminate malaria within relatively short time frames have to address the recurrent nature of vivax infections in areas where G6DPdeficiencies are common but vary widely in nature. Vivax also occurs now mainly in remote and mobile and migrant populations which require management by lower level health workers so the aim to develop a universal gold standard against which to measure any new tests is important. With the introduction of the only new drug to be developed to address the presence of hypnozoites this is even more urgent.

The number of studies and the range of countries where studies were done give a range of countries with different prevalences and varieties of G6DPd but did not include data from countries with very high rates and severe varieties such as Pakistan.

In light of the fact that universal testing of newborns for G6DP is being suggested and indeed implemented in some countries the important variation in sensitivity and specificity of the tests in newborns is an important area to explore further. 

Although patients with and without malaria were looked at it might have been interesting to know if other blood dyscrasias common in these populations such as thalassemia and sickle cell disease had an influence on the outcome of the tests if any 

The use of the term gold standard seems a little ambitious as the standard cannot be a universal standard if the standard has to be set for the different assays and in different populations as seems to have to be the case. The intra-laboratory variation as well as the variation across different assay types requires that quality control is especially important and this is a very useful fact to emphasise especially in countries where quality control systems are weak.

When introducing the drugs in line 162 It might have been a good idea to explain the different ways these drugs are deployed ie Primaquine is used as a single low dose to deal with gametocytes of Pf which WHO says is safe for all levels of G6DP activity (although in countries with very weak pharmacovigilance systems this contention may not be completely true) Primaquine for 14 or 7 days (or over 8 weeks) for management of hypnozoites and tafenoquine which is a single dose and therefore carries more hazards and requires a higher degree of safety measures before administration.

When relying on qualitative tests ( those are the only ones universally available at the moment to National programmes the sensitivity and specificity of these tests against a locally derived gold standard is important Even when Primaquine is in the national protocol many health workers are reluctant to prescribe this drug due to perhaps exaggerated fears of Haemolysis. However good a test is it can never be perfect and so cases of haemolysis will occur and when training health workers and drug recipients it will be necessary to advise patient of how to detect haemolysis and provide access to transfusion services when needed. This will be even more important when tafenoquine is used as management of drug induced haemolysis will be even more challenging

[LINK]

---

## [Decision Letter · Decision Letter 1]

10 Feb 2020

Dear Dr. Pfeffer,

Thank you very much for re-submitting your manuscript "Quantification of glucose-6-phosphate dehydrogenase activity by spectrophotometry: A systematic review and meta-analysis" (PMEDICINE-D-19-02590R1) for consideration at PLOS Medicine.

I have discussed the paper with editorial colleagues and our academic editor and it was also seen again by one reviewer. I am pleased to tell you that, provided the remaining editorial and production issues are dealt with, we expect to be able to accept the paper for publication in the journal.

[LINK]

Please let me know if you have any questions. Otherwise, we look forward to receiving your revised manuscript shortly. 

Sincerely,

Richard Turner PhD, for Thomas McBride, PhD

rturner@plos.org

Requests from Editors:

To your abstract, please add summary demographic details for study participants. 

In your abstract and elsewhere, please add p values alongside 95% CI where available. 

At line 122, would "... lack of availability of covariate data ..." be more appropriate?

At line 124, please begin the sentence "Our findings indicate that ..." or similar.

If appropriate, please adapt the wording at the start of the Methods section of your main text (around line 239 where you refer to the PROSPERO record) to note that the study was prospectively planned 

Please refer to the attached PRISMA checklist in the Methods section.

Towards the end of the Methods section, please add a sentence to note that specific ethics approval was not required for the present analysis, and if available the identity of the body responsible for that assessment.

At line 305, please adapt the text "a total of ... sample data" to indicate whether this refers to number of individuals.

We suspect that the reference call-outs in table 2 should not be in superscript.

Please adapt reference call-outs to this style: "... endemic regions [3,4].".

Please add full access details to reference 7. 

Comments from Academic Editor:

This is an interesting, pertinent, and well conducted study in an area with high levels of interest in malaria and concomitant funding. This analysis provides one part of the picture in moving forward. It’s clearly written and relatively short. I have only minor comments about language in a couple of places.

I agree with previous comments of referees that the authors should not refer to spectrophotometry as the “Gold standard”; it is not, this is a notoriously difficult condition to measure, there are so many variants, and the functional significance, in relation to the extent of the haemolysis with with tafenoquine, is really not clear. Thus “reference standard” should be used throughout.

The authors conclude that there is “substantial heterogeneity in G6PD measurements between sites…” I would encourage they describe this as “substantial variation in….”. The word heterogeneity implies that G6PD actually does vary sites but in fact it appears that it is the result largely of laboratory measures varying. 

Under, what did the researchers find, third bullet, please try and avoid the use of the word “significant”. This is an ambiguous word, it can be qualitatively important, or it can mean some statistical evaluation using P 0.05. “Important” or “considerable” is more helpful.

On line 189, the authors don’t quite communicate to the reader that the concern with tafenoquine is the long half-life, and thus the persistence of the harmful haemolytic effect with this drug-it may be worse than PQ.

Otherwise the message is clear

Comments from Reviewers:

*** Reviewer #2: 

The revised version of the manuscript is much improved. The authors have responded appropriately the reviewers comments and have substantiated their responses well. The methods are clearer now related to the pooling process and the authors have described the limitations of obtaining study-level covariates to explore heterogeneity. Thank you for also producing summary ROC curves in supplemental, stratified by gender and threshold. On this basis, my recommendation would for acceptance for publication, given two minor points:

1) Thank you for the detailed response to the IPD methodology. The authors mention that "had we adjusted our pooled threshold with the covariates available, this would have meant that practitioners would also have to adjust their assay cut-offs based upon local factors - a level of complexity which is unlikely to be practical or offer utility beyond current practice." This is an interesting point and I thought warrants inclusion in the methodology to strengthen their rationale. 

2) I thank the authors for educating me on the WHO guidelines being out of date - I think this also is a nice point that should have a place in the discussion - The policy implications are that both thresholds being inadequate.

***

[LINK]

---

## [Editor Report · Decision Letter 2]

13 Apr 2020

Dear Dr. Pfeffer, 

On behalf of my colleagues and the academic editor, Dr. Paul Garner, I am delighted to inform you that your manuscript entitled "Quantification of glucose-6-phosphate dehydrogenase activity by spectrophotometry: A systematic review and meta-analysis" (PMEDICINE-D-19-02590R2) has been accepted for publication in PLOS Medicine. 

PRODUCTION PROCESS

PRESS

PROFILE INFORMATION

Thank you again for submitting the manuscript to PLOS Medicine. We look forward to publishing it. 

Best wishes, 

Thomas McBride, PhD

Senior Editor 

PLOS Medicine

plosmedicine.org